# Measuring Nasal Patency and the Sense of Smell in CRSwNP Patients Treated with Dupilumab

**DOI:** 10.3390/jpm13020234

**Published:** 2023-01-28

**Authors:** Giancarlo Ottaviano, Eugenio De Corso, Elena Cantone, Andrea Ciofalo, Tommaso Saccardo, Riccardo Bernardi, Edoardo Mairani, Claudio Montuori, Giuseppe Roccuzzo, Livia Soldati, Benedetto Randon, Sonny Zampollo, Alessandra Di Chicco, Francesca Visconti, Bruno Scarpa, Piero Nicolai

**Affiliations:** 1Otolaryngology Section, Department of Neurosciences DNS, University of Padova, 35128 Padova, Italy; 2ENT Department of A. Gemelli Unversitary Hospital IRCCS, 00168 Rome, Italy; 3Reproductive and Odontostomatological Sciences-ENT Section, Department of Neuroscience, University of Naples Federico II, 80131 Naples, Italy; 4Rhinology Unit, Department of Sensory Organs, Sapienza University of Rome, 00161 Rome, Italy; 5Department of Statistical Sciences, University of Padova, 35100 Padova, Italy

**Keywords:** dupilumab, PNIF, SNOT-22, VAS, NPS, Sniffin’ Sticks, PROMs

## Abstract

Chronic rhinosinusitis with nasal polyps (CRSwNP) in the most severe forms is associated with a poor quality of life. Dupilumab has been suggested as an add-on treatment option for severe CRSwNP. Severe CRSwNP patients treated with dupilumab in different rhinological units were followed up at 1, 3, 6 and 12 months from the first administration and were considered for this study. At baseline (T0) and at each follow-up, patients underwent nasal endoscopy and completed the sinonasal outcome test (SNOT)-22, a visual analogue scale (VAS) for smell/nasal obstruction, peak nasal inspiratory flow (PNIF) and the Sniffin’ Sticks identification test (SSIT). The aim of the present study was to evaluate the effects of dupilumab in patients with severe uncontrolled CRSwNP on recovering nasal obstruction and smell impairment. Moreover, the method between PNIF and SSIT with the highest correlation with patients’ response to dupilumab was evaluated. One hundred forty-seven patients were included. All parameters improved during treatment (*p* < 0.001). At T0, no correlations were found between PNIF and nasal symptoms. Nevertheless, during the following evaluations significant correlations between PNIF changes and both nasal symptoms and NPS were observed (*p* < 0.05). At T0, SSIT did not correlate with SNOT-22. Similarly to PNIF, during the follow-up SSIT changes significantly correlated with nasal symptom and NPS (*p* < 0.05). Comparing PNIF and SSIT correlations with SNOT-22 and NPS, PNIF showed a higher correlation with both. Dupilumab improves nasal obstruction and the sense of smell. PNIF and SSIT are effective tools in monitoring patients’ response to dupilumab.

## 1. Introduction

Chronic rhinosinusitis with nasal polyps (CRSwNP) is a multifactorial disease that significantly affects patients’ quality of life (QoL) [1]. It can be associated with genetic disorders, immunodeficiency, atopy, anatomical abnormalities and chronic osteitis, but can also be influenced by exposure to environmental factors such as air pollution, smoke, viruses, bacteria, fungi and allergens [2]. In particular, atopy may not contribute directly to the pathogenesis of chronic rhinosinusitis (CRS) and could be a coexisting disease, but it may also play a disease-modifying role in CRS. In fact, atopy seems to be a possible useful prognostic factor for the predisposition to relapse after endoscopic sinus surgery (ESS). Atopy is defined as a predisposition to react immunologically to a variety of antigens and allergens, leading to CD4+ Th2 differentiation and overproduction of immunoglobulin E (IgE). The most common clinical consequences are allergic bronchial asthma and allergic rhinitis, followed by atopic dermatitis and food allergy. More than one clinical disease can coexist in an individual at the same time or at different times [1,2].

In the last 2–3 years, due to the improvement in our knowledge on the underlying pathophysiological mechanisms, we experienced a crucial change in the approach and management of CRSwNP, mainly related to the new therapeutic opportunities like targeted biologic molecules. This new approach is based on the identification of patients that have a higher chance to respond to a specific treatment according to the inflammatory pathways of the disease, following the principles of so-called precision medicine [3]. In this scenario, dupilumab (a biologic drug able to simultaneously block both IL-4 and IL-13 pathways) has been demonstrated effective in reducing the size of the nasal polyps and improving their QoL not only in randomised clinical trials [3], but also in real-life studies conducted on severe CRSwNP patients [4,5,6].

Nasal obstruction is one of the main symptoms reported by CRS patients. In particular, more than 90% of CRSwNP patients complain of nasal blockage [7,8]. Nasal obstruction can be measured subjectively using some patient reported outcome measures (PROMs) such as the visual analogue scale for nasal obstruction (VAS-NO). Nevertheless, literature data suggest that the correlation between the subjective sensation of nasal obstruction and its objective measurement can be unpredictable [9,10]. This leads to the need of reliable objective instruments for the measurement of nasal obstruction [11], especially in the era of biologicals. In clinical practice, rhinomanometry (RM), acoustic rhinometry (AR) and peak nasal inspiratory flow (PNIF) are among the most popular tools used to objectively assess nasal obstruction [11]. The latter consists of a face mask which the patient applies over the nose (without touching it) with the mouth closed. The patient sniffs air through the nose and the peak flow is recorded by a cursor [12]. Different studies have compared PNIF with other objective methods for the measurement of nasal obstruction, finding a correlation among them [13,14]. In particular, a very recent study demonstrated PNIF to better correlate with the patients’ perception of nasal obstruction than both AR and four-phase RM [15].

Likely, the subjective perception of olfactory impairment (i.e., using VAS for smell) may lead to poor or absent correlation with the results of olfactory tests [16,17], so a smell test should always be associated with a PROM for smell in order to better define the presence of dysosmia and its severity [18,19,20].

The aim of the present multicentric study was to evaluate the effects of dupilumab in patients with severe uncontrolled CRSwNP on recovering nasal obstruction and smell impairment measured, respectively, by PNIF and the Sniffin’ Sticks identification subtest (SSIT). Furthermore, the method between PNIF and SSIT with the highest correlation with patients’ response to dupilumab (in terms of SNOT-22 and NPS reduction) was evaluated.

## 2. Materials and Methods

### 2.1. Population

This is a no profit, observational retrospective multicentric study conducted including four different ENT Italian centres. Patients were recruited at the Rhinological Unit of Padua University Hospital, Rhinology Unit of the Gemelli Hospital Foundation–IRCCS Catholic University of Sacred Heart of Rome Sapienza University of Rome and University Federico II of Naples. The patients were treated and followed between February 2020 and November 2022.

All the patients enrolled in the study were older than 18 years, suffering of severe CRSwNP, defined by nasal polyp score (NPS) ≥5 and/or a sinonasal outcome test-22 (SNOT-22) ≥50, with inadequate symptom control despite INCS use, receiving at least 2 cycles of systemic corticosteroid in the last year, or having had sinonasal surgery (ESS). Dupilumab was administered subcutaneously 300 mg every 2 weeks as an add-on therapy to intranasal corticosteroids (INCS). The exclusion criteria were pregnancy, radiochemotherapy for cancer in the 12 months before the start of the treatment, or concomitant long-term oral corticosteroid therapy for chronic autoimmune disorders. The study was conducted in accordance with the 1996 Helsinki Declaration and was approved by the hospital ethical committees (Padua University Hospital: 5304/AO/22; Gemelli Hospital of Rome: ID 4429; University Federico II of Naples: 424/21; Sapienza University of Rome: 0329/2022). Informed consent was obtained from each subject before starting any study-related procedure.

### 2.2. Clinical Evaluation

We collected data at baseline (before starting the biological treatment) (T0) and at the following follow-up visits (1 month (T1), 3 months (T2), 6 months (T3) and 12 months (T4)). At baseline and at each follow-up, patients underwent nasal endoscopy using a 0° and/or 30° rigid endoscope and the NPS score was calculated for all of them according to Gevaert et al. [21]. QoL was evaluated using the SNOT-22 questionnaire [19]. Nasal obstruction and impairment of sense of smell were measured subjectively by VAS-smell and VAS-NO [19]. Nasal airflow was assessed by means of PNIF (Clement Clark International, Harlow, UK) [10] while olfaction by means of SSIT (16 odours) (Burghart Messtechnik GmbH, Holm, Germany) [20].

### 2.3. Statistical Analysis

Sample quantiles were used to describe the effect of all relevant variables in time and Bravais–Pearson correlation coefficient was used to measure the relations between the different indicators. Groups of postsurgical and naïve patients were compared with Wilcoxon test for quantitative variables and with the Fisher exact test for the qualitative ones. For all tests, *p*-values were calculated, and 5% was considered as the critical level of significance.

Mixed effect models were fitted to data to take into account their longitudinal characteristic. Following previous work [12], we reduced the heterogeneity in variability of PNIF by taking the transformation MODPNIF = (PNIF)^1/2^. Given the nonlinearity of all the effects in time, a parameter for each time contrast was included in the model and all significances of the effects of time and all the other variables were obtained by ANOVA tables with F-tests and *p*-values obtained using Satterthwaite’s method [22] for denominator degrees-of-freedom and F-statistic. The best multiple predictive models for SNOT-22 and NPS were selected via backward stepwise selection based on the Akaike information criterion (AIC) [23]. AIC was also used for comparing the different models. A better model is characterised by a lower AIC value.

The R: a language and environment for statistical computing (R Foundation for Statistical Computing, Vienna, Austria) was used for all analyses [24].

## 3. Results

A total of 147 patients (97 males and 50 females, mean age 51.6 ± 13.9 years) undergoing dupilumab as add-on therapy to INCS for at least 1 year were considered for the present multicentric study. Patients’ main clinical characteristics at baseline (T0) are reported in Table 1. During the study period, three patients complained of mild arthralgias that had rapid and spontaneous resolution (within one–two weeks) and two patients discontinued dupilumab for severe arthralgia, in one patient associated with hyperosinophilia (>1500 UI/L). Finally, two patients experienced periorbital edema, which resolved well after medical therapy.

Table 2 shows differences in scores for patient reported outcome measures (PROMs) (SNOT-22, VAS scores for smell and NO, NCS and ACT) and objective measurements (NPS, PNIF, SSIT and cytology findings) during the follow-up.

SNOT-22, VAS-smell, VAS-NO and NPS significantly decreased through the study period (*p* < 0.0001). Conversely, PNIF showed a significant increase at the first follow-up control (T1) with respect to the baseline (T0) and maintained its improvement during the 12 months follow-up (T2, T3 and T4) (*p* < 0.0001). Although at the baseline (T0) there was a significant correlation between PNIF and NPS (*p* < 0.0001), no correlations could be found between PNIF and symptoms (VAS-NO, SNOT-22 and VAS-smell). When looking at PNIF changes during the follow-up with the statistical model, significant negative correlations between PNIF and both nasal symptom and NPS were observed (PNIF and SNOT-22 (*p* = 0.0242), Figure 1a; PNIF and VAS-NO (*p* <0.0001), Figure 1b; PNIF and VAS-smell (*p* =0.00342), Figure 1c; PNIF and NPS (*p* <0.0001), Figure 1d).

SSIT improved significantly after the baseline (T0) and maintained its improvement through the 12 months follow-up (T1, T2, T3, T4) (*p* < 0.0001). At T0, a correlation was observed between SSIT and VAS-smell (*p* < 0.001), VAS-NO (*p* = 0.02) and NPS (*p* = 0.012), but no correlations were observed with SNOT-22. When evaluating SSIT changes during the follow-up through the statistical model, negative significant correlations between SSIT and both nasal symptoms and NPS were found (SSIT and SNOT-22 (*p* = 0.0002), Figure 2a; SSIT and VAS-smell (*p* < 0.0001), Figure 2b; SSIT and VAS-NO (*p* < 0.0001), Figure 2c; SSIT and NPS (*p* = 0.03), Figure 2d).

The same results regarding PNIF and SSIT were obtained in a multivariate analysis, which considered also the effects of age, sex, smoke, atopy, nonsteroidal anti-inflammatory drugs (NSAIDs) intolerance, previous ESS, the number of systemic corticosteroids cycles taken in the last year and the presence of asthma.

Considering the AIC information criterion to find the variable between PNIF and SSIT with the highest correlation with nasal symptoms (SNOT-22), given the effect of time, PNIF showed a better correlation with SNOT-22 reduction due to dupilumab (AIC PNIF = 5843.9, AIC SSIT = 5887.6). Considering PNIF and SSIT together, we observed a higher correlation with SNOT-22 reduction after starting dupilumab than with the single variables (either PNIF or SSIT) (AIC = 5834.3).

Considering the AIC information criteriona find the variable between PNIF and SSIT with the highest correlation with NPS, given the effect of time, the lower value was obtained by PNIF, which therefore showed its better correlation with NPS reduction due to dupilumab (AIC PNIF = 2526.9, AIC SSIT = 2558.3). Once considering both PNIF and SSIT together, the correlation was similar to that of only PNIF (AIC = 2531.4).

## 4. Discussion

Dupilumab is approved for five indications: moderate-to-severe atopic dermatitis, moderate-to-severe eosinophilic or OCS-dependent asthma, eosinophilic esophagitis, prurigo nodularis and severe uncontrolled CRSwNP. In the present multicentric study, dupilumab showed to be effective in the treatment of severe and uncontrolled CRSwNP. Since the first follow-up visit (T1), both PNIF and SSIT showed a significant improvement when compared to T0. Similarly, SNOT-22, VAS-NO, VAS-smell and NPS significantly improved at T1 and kept this trend through the follow-up.

Although at T0 no correlations were observed between PNIF/SSIT and SNOT-22, considering PNIF/SSIT changes during the follow-up both these measures showed to significantly correlate with PROMS and vice versa, independently of the major phenotyping variables that can affect CRSwNP severity (age, sex, smoke, atopy, NSAIDs intolerance, previous ESS, number of systemic corticosteroids cycles taken and asthma). A similar result was already shown in a very recent real-life observational study conducted to evaluate the efficacy of dupilumab in patients with uncontrolled CRSwNP. The authors observed a general moderate correlation between PNIF and VAS-NO and between SSIT and VAS-smell especially during the follow-up and concluded that once the sinonasal inflammation is under control, the patients are able to give a better and more reliable evaluation of their nasal symptoms [5]. At T0 and during the follow-up, both PNIF and SSIT correlated with NPS.

PNIF showed a higher correlation than SSIT with nasal symptoms (SNOT-22 reduction), proving to be a more effective method to evaluate the response to dupilumab in severe uncontrolled CRSwNP patients. This result seems to confirm the observations of Abdala et al. who demonstrated that nasal blockage was the most reported symptom in CRSwNP patients as measured by means of SNOT-22 (93.5% of the patients), followed by the altered sense of smell (75.7% of the patients) [7]. Nasal obstruction and dysosmia are the most reported complaints by these patients. It is not surprising that the correlation of PNIF and SSIT together on SNOT-22 reduction was higher than the one of PNIF considered on his own. According to these results, the measurement of nasal obstruction (i.e., using PNIF) and sense of smell (i.e, using SSIT) should be considered of utmost importance as it could give more information than measuring only one of these two variables. Considering the correlations of PNIF and SSIT on NPS, once again PNIF showed to have a higher correlation than SSIT on NPS reduction. Furthermore, PNIF showed to have the same correlation on NPS reduction than that of PNIF and SSIT together. This result seems to demonstrate that NPS reduction produces a clear improvement of nasal flows due to a volumetric reduction in the polyps. Nevertheless, nasal polyps’ reduction seems to be associated with improved sense of smell to a lesser degree in these patients. A very recent study demonstrated that there was no correlation between NPS and SSIT or VAS-smell in severe CRSwNP patients treated with dupilumab, suggesting that the sense of smell (measured by SSIT) does not depend only on the volume of polyps and may be mainly consequence of the reduced inflammation of the sinonasal mucosa [25]. Future studies, including the study of inflammatory biomarkers, may be useful to clarify this aspect.

The impact of therapy with dupilumab on severe uncontrolled CRSwNP patients is clinically remarkable. In this multicentric study, short-term (1 month) and long-term (1 year) results have shown a significant improvement in patients’ QoL. The use of simple tests such as PNIF and SSIT can significantly help to both phenotype the severe forms of CRSwNP treated with biologics and monitor the response to treatment during the follow-up. Unfortunately, although nasal obstruction is one of the major symptoms reported by these patients and it can be underestimated by PROMS, nasal patency is only seldom objectively evaluated in clinical practice. As shown in this study, PNIF correlates with both SNOT-22 and NPS reduction in CRSwNP patients treated with dupilumab, so it could be certainly useful in clinical practice. There is emerging evidence in the literature that CRSwNP patients undergoing dupilumab tend to overestimate smell function improvements on PROMS compared to objective valuation with SSIT [25]. Despite international guidelines emphasising the importance of olfactory assessment [26], in everyday clinical practice olfactory tests are still considered nonessential (i.e., before starting biological treatments) by a significant percentage of ENT colleagues, as these tests are often not available in outpatient clinics. The present study shows a correlation between SSIT and SNOT-22 reduction in CRSwNP patients treated with dupilumab, pointing toward the utility of this quick and simple olfactory test in clinical practice.

The main limitations of this study are its retrospective nature, the relatively short follow-up and the lack of laboratory data to describe the immunologic status of the population. Otherwise, the strengths are its multicentric nature, the numerosity of the analysed population, the follow-up design shared between the four ENT centres, the use of validated tests as evaluation tools and that patients in each centre were evaluated by the same techniques.

## 5. Conclusions

Severe sinonasal inflammation is associated with nasal obstruction and olfactory loss in patients affected by uncontrolled CRSwNP. Although PNIF alone does not seem to correlate with sinonasal symptoms, assessing its trend through time, together with SSIT (16 odours test), seems to be effective in monitoring patients’ response to the biologic-targeted therapy with dupilumab. These objective, rapid, simple, inexpensive and easy-to-perform methods to measure nasal patency and olfaction can be helpful before and during follow-up of these patients. PNIF and SSIT, in association with PROMS, are crucial for a high-level phenotyping process to better assess and monitor the response to high-cost biological treatments.

## Figures and Tables

**Figure 1 jpm-13-00234-f001:**
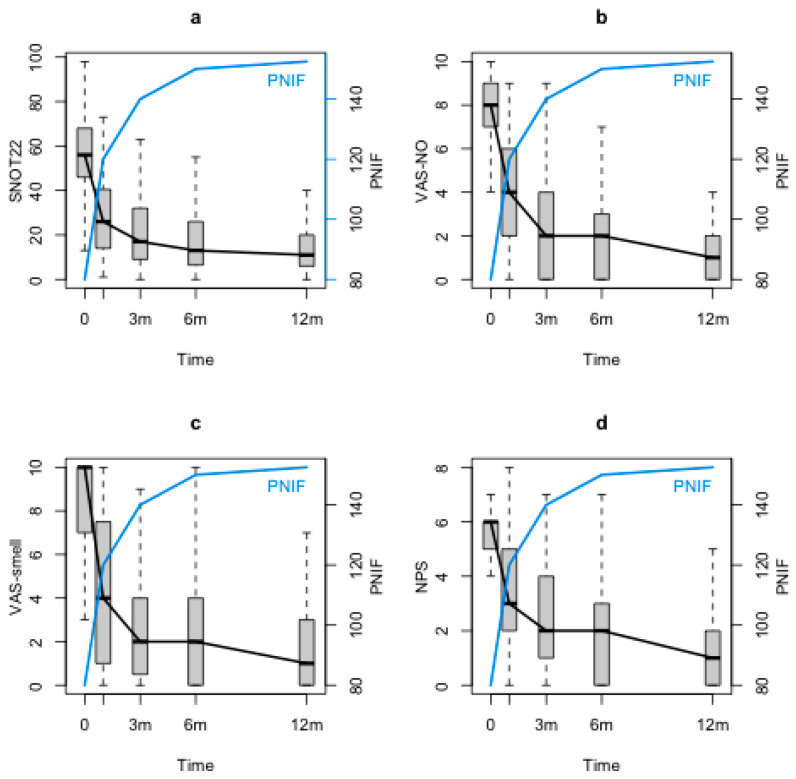
PNIF, SNOT-22, VAS-NO, VAS-smell and NPS changes during the follow-up. PNIF: peak nasal inspiratory flow, SNOT-22: sinonasal outcome test-22; VAS-NO: visual analogue scale for nasal obstruction; VAS-smell: visual analogue scale for smell; NPS: nasal polyps score; m: months.

**Figure 2 jpm-13-00234-f002:**
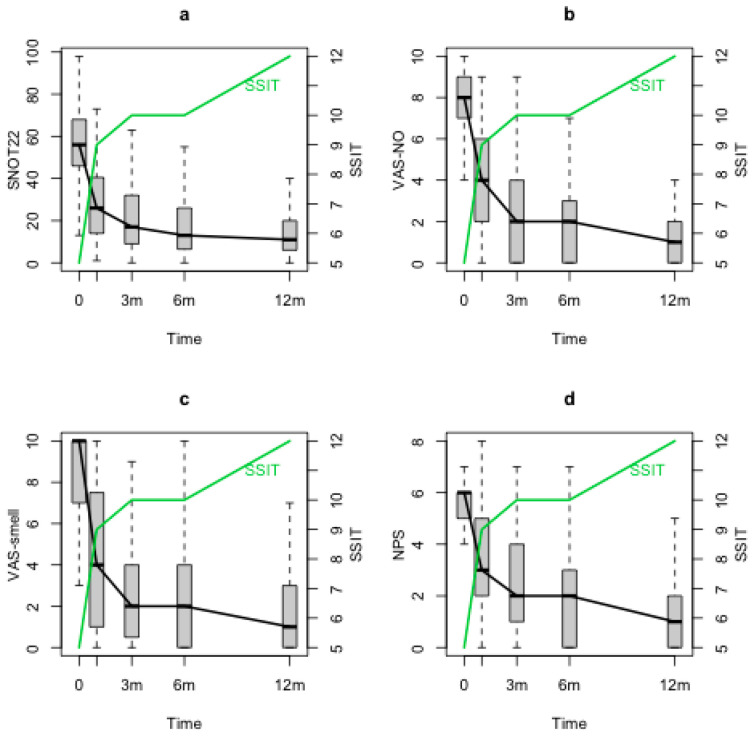
SSIT, SNOT-22, VAS-NO, VAS-smell and NPS changes during the follow-up. SSIT: Sniffin’ Sticks identification test (16 odours), SNOT-22: sinonasal outcome test-22; VAS-NO: visual analogue scale for nasal obstruction; VAS-smell: visual analogue scale for smell; NPS: nasal polyps score; m: months.

**Table 1 jpm-13-00234-t001:** Patients’ main clinical characteristics at baseline.

	ALL *n* = 147	Padua *n* = 45	Rome, Gemelli *n* = 51	Rome, Sapienza *n* = 28	Napoli, Federico II *n* = 23
Sex	50 Women 97 Men	11 Women 34 Men	21 Women 30 Men	9 Women 19 Men	9 Women 14 Men
Mean Age, yr (SD)	51.6 (13.9)	51.6 (13.4)	50.6 (13.4)	49.7 (13.3)	55.6 (16.87)
Asthma, n (%)	92 (63)	24 (53)	36 (71)	11 (39)	21 (91)
NSAIDs intolerance, n (%)	35 (24)	10 (22)	16 (31)	4 (14)	5 (22)
Widal triad, n (%)	31 (21)	9 (20)	14 (27)	3 (10)	5 (22)
Allergy, n (%)	86 ()	34 (75)	26 (51)	6 (21)	20 (87)
Smokers, n (%)	32 (22)	15 (33)	10 (20)	4 (14)	3 (13)
Previous ESS, n (%)	133 (90)	42 (93)	49 (96)	23 (82)	19 (83)
Mean n of previous surgeries, n (SD)	1.61 (1.06)	1.7 (0.93)	1.92 (0.89)	1.28 (1.56)	1.08 (0.59)

SD: standard deviation; NSAIDs: nonsteroidal anti-inflammatory drugs; OCS: oral corticosteroids; ESS: endoscopic sinus surgery.

**Table 2 jpm-13-00234-t002:** Patients’ clinical changes during the follow-up.

	T1 vs. T0		T2 vs. T0		T3 vs. T0		T4 vs. T0	
	Difference	*p*	Difference	*p*	Difference	*p*	Difference	*p*
VAS NO	−3.84	<0.001	−5.25	<0.001	−5.52	<0.001	−6.10	<0.001
VAS smell	−3.76	<0.001	−5.22	<0.001	−5.53	<0.001	−6.04	<0.001
SNOT22	−28.45	<0.001	−34.96	<0.001	−38.47	<0.001	−42.15	<0.001
NPS	−2.35	<0.001	−3.02	<0.001	−3.05	<0.001	−4.03	<0.001
PNIF	34.83	<0.001	45.55	<0.001	57.21	<0.001	63.49	<0.001
SSIT	3.20	<0.001	3.76	<0.001	4.14	<0.001	5.18	<0.001

VAS-NO: visual analogue scale for nasal obstruction; VAS-smell: visual analogue scale for smell; SNOT-22: sinonasal outcome test-22; NPS: nasal polyp score; PNIF: peak nasal inspiratory flow; SSIT: Sniffin’ Sticks identification test; *p*: *p*-value. T0: baseline; T1: 1 month after the first dupilumab administration; T2: 3 months after first dupilumab administration; T3: 6 months after first dupilumab administration; T4: 12 months after first dupilumab administration.

## Data Availability

The datasets generated and analysed during the current study are available on reasonable request.

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
