# Peer review of "Measuring Nasal Patency and the Sense of Smell in CRSwNP Patients Treated with Dupilumab"

_jpm, 2023, doi:10.3390/jpm13020234_

Round 1
Reviewer 1 Report
It is a clinically importnat study, nicely performed, with good and elaborated methodology.
I have few text remarks:
1. Introduction –line 35- Factors that may contribute to CRSwNP. As much as I know there is no bone marrow in the sinuses, so I would correct chronic osteomyelitis to chronic osteitis.
2. Methods -line 110 - Please erase duplicate sentence
3. Results - line 123-124- add on therpy ...to what ? INCS (Intranasal corticosteroids) ?
Questions :
1.I expected the authors to conclude that an absolute result of the PNIF alone dose not correlate with symptoms but the trend or change over time ( after surgery or medical tx as dupilumab) is significant.
2. Was Dupilumab 300mg s.c was given twice or once a month ? please add information.
3. we don't have laboratory data of the study population (peripheral eosinophils , total IGE) so we can clearly identify the heterogenicity of this population (type 2 or non type 2 population). - if possible please add.
Author Response
REV.1
Editor
Personalized Medicine
Padova, 15/1/2023
Dear Editor,
We are submitting the revised manuscript “Measuring nasal patency and the sense of smell in CRSwNP patients treated with DUPILUMAB” which was edited according to reviewers’ comments and we would like you to consider for publication in “Journal Personalized Medicine” for the Special Issue “The Challenges and Prospects in Diagnostics of Otolaryngology" (guest editor, Prof. Elena Cantone)”.
We would like to thank the reviewer panel for reconsidering of the article.
The changes in the manuscript have been highlighted and a list of all changes with a point-by-point reply to the reviewer comments is enclosed.
Best regards,
Giancarlo Ottaviano, MD, PhD
Correspondence to: Giancarlo Ottaviano, MD, PhD, Dept. Neurosciences, Otolaryngology Section, Padova University, Italy; email: giancarlo.ottaviano@unipd.it; Fax +39 049 8213113; Tel +39 049 8212029
According to reviewer 1 suggestions:
- Introduction –line 35- Factors that may contribute to CRSwNP. As much as I know there is no bone marrow in the sinuses, so I would correct chronic osteomyelitis to chronic osteitis“. Thank you for this suggestion. We amended in the text (page 2, line 39)
- Methods -line 110 - Please erase duplicate sentence”. Thank you for this suggestion. We erased the duplicated sentence.
- Results - line 123-124- add on therpy ...to what ? INCS (Intranasal corticosteroids) ?”. Thank you for this suggestion. Yes, it was an add on therapy to the standard of care which are INCS. We added the information to the manuscript (line 136).
- Question number 1 “I expected the authors to conclude that an absolute result of the PNIF alone dose not correlate with symptoms but the trend or change over time (after surgery or medical tx as dupilumab) is significant.”Thank you for pointing it out, we included this aspect in the “Conclusion” paragraph (266-267)
- Question number 2. “Was Dupilumab 300mg s.c given twice or once a month? please add information”. Dupilumab was administrated once every 15 days, we added the information as requested (lines 97-98)
- Question number 3. “we don't have laboratory data of the study population (peripheral eosinophils , total IGE) so we can clearly identify the heterogenicity of this population (type 2 or non type 2 population). - if possible please add”. We agree with the reviewer for this remarkable suggestion. Unfortunately, we do not have enough data to clarify that, as this aspect was not within the aims of the study. In future studies we will take in consideration this aspect. We have discussed this limitation of the study in the discussion section (line 258-263).

Reviewer 2 Report
The authors reported the results of a multicenter study with the aim of assessing the effects of dupilumab in patients with uncontrolled CRSwNP. The manuscript is interesting and well-written. I have few comments.
My comments:
- Abstract: abstract should be unstructured
- Abstract: the aim of the study should be reported in the abstract
- Introduction: in the introduction the concept of atopy (atopic dermatitis and its phenotypes, allergic rhinitis,..) should be discussed. Please read and cite " doi: 10.1097/ACI.0000000000000837"
- Material and Methods: please report the dosage of dupilumab
- Results: safety data should be reported
- Discussion:: dupilumab showed to be effective and safe in patients with AD and atopic dermatitis or in other atopic diseases. Please read and cite "doi: 10.1111/dth.15120" and "doi: 10.1080/09546634.2022.2102121"
- Strengths and limitations: strengths and limitations of the study should be discussed in a separate section
- A table summarizing the results in terms of efficacy at each time point with their statistical significance should be reported
Author Response
REV.1
Editor
Personalized Medicine
Padova, 15/1/2023
Dear Editor,
We are submitting the revised manuscript “Measuring nasal patency and the sense of smell in CRSwNP patients treated with DUPILUMAB” which was edited according to reviewers’ comments and we would like you to consider for publication in “Journal Personalized Medicine” for the Special Issue “The Challenges and Prospects in Diagnostics of Otolaryngology" (guest editor, Prof. Elena Cantone)”.
We would like to thank the reviewer panel for reconsidering of the article.
The changes in the manuscript have been highlighted and a list of all changes with a point-by-point reply to the reviewer comments is enclosed.
Best regards,
Giancarlo Ottaviano, MD, PhD
Correspondence to: Giancarlo Ottaviano, MD, PhD, Dept. Neurosciences, Otolaryngology Section, Padova University, Italy; email: giancarlo.ottaviano@unipd.it; Fax +39 049 8213113; Tel +39 049 8212029
According to reviewer 2 suggestions:
- Question number 1. “Abstract: abstract should be unstructured”.
We made the adjustment as requested.
- Question number 2. “Abstract: the aim of the study should be reported in the abstract”.
Thank you for this suggestion. We added the aims of the study in the abstract (lines 22-25).
- Question number 3. “Introduction: in the introduction the concept of atopy (atopic dermatitis and itsphenotypes, allergic rhinitis,..) should be discussed”.
We made that adjustment as requested (41-49).
- Question number 4. “Material and Methods: please report the dosage of dupilumab”.
We added this information as requested (line 97).
- Question number 5 “Results: safety data should be reported”. Thank you for this suggestion. We observed some adverse events during the study and we added this information in the text (lines 138-142).
- Question number 6 “Discussion: dupilumab showed to be effective and safe in patients with AD and atopic dermatitis or in other atopic diseases”. Thank you for this suggestion. We added this information reporting the indications of dupilumab also in other diseases (lines 199-201).
- Question number 7. “Strengths and limitations: strengths and limitations of the study should be discussed in a separate section”. Thank you for pointing it out, in the discussion section we included the strengths and limitations of the study (258-263).
- Question number 8. “A table summarizing the results in terms of efficacy at each time point with their statistical significance should be reported”.
Thank you for this suggestion. We have added a table (table 2) showing the efficacy of dupilumab at each time point for the variables studied.
